

# Daytime driving decreases amphibian roadkill

Wenyan Zhang[1,2,*], Guocheng Shu[1,2,*], Yulong Li[1,2], Shan Xiong[1,2], Chunping Liang[3] and Cheng Li[1]

[1] Chengdu Institute of Biology, Chinese Academy of Sciences, Chengdu, China
[2] University of Chinese Academy of Sciences, Beijing, China
[3] Wanglang National Nature Reserve, Mianyang, China
* These authors contributed equally to this work.

## ABSTRACT

Roadkill has gradually become a common factor that has contributed to the decline of amphibians, and traffic volume is an important parameter that can be used to determine the impacts of roads. However, few researchers have studied the effects of either daily or nightly traffic volume on amphibian roadkill in China. Hence, as an essential step for implementing mitigation measures, we conducted 77 road surveys along 10 km of road in the Wanglang National Nature Reserve (NNR) to determine the temporal and spatial distribution patterns of amphibian road mortality. In total, 298 dead individuals (*Bufo andrewsi* and *Rana chensinensis*) were observed on the road from April to October in 2017 and during June and August in 2015 and 2016. *B. andrewsi* had the highest number of records (85.2%) and was more vulnerable to road mortality than *R. chensinensis*. Amphibian fatalities mainly occurred during the breeding season in April, but there was an additional concentration of *B. andrewsi* roadkill in June and July. There was a significantly positive correlation between amphibian road mortality and mean night-time traffic volume. Roadkill hotspots were non-randomly distributed throughout the study area and were mainly concentrated in the road sections near the breeding pools. Therefore, to effectively mitigate the effects of road mortality in the Wanglang NNR, measures should be implemented both during hot moments and at hotspots. First, based on roadkill hot moments, during the breeding season (in April) and in June and July, the Wanglang NNR should establish temporary traffic restraints at night. Second, based on roadkill hotspots, culverts should be established in areas near breeding pools adjacent to roads, and barrier walls should be installed to guide amphibians into the culverts.

## INTRODUCTION

Roads lead to the deaths of vertebrates due to collisions with vehicles, and roadkill events have long been considered to have a more serious effect than hunting on vertebrate mortality in the USA (*Forman & Alexander, 1998*). In particular, collision with vehicles is a major cause of amphibian mortality (*Hels & Buchwald, 2001*; *Puky, 2005*; *Cairo & Zalba, 2007*; *Glista, DeVault & DeWoody, 2008*). Moreover, amphibians are one of

Corresponding author
Cheng Li, licheng@cib.ac.cn

the most threatened vertebrate groups (*Stuart et al., 2004*), with nearly 40% of living amphibian species in danger of extinction (*IUCN, 2011*). Amphibians have several unique life-history or behavioural traits, such as seasonal migrations (e.g., from overwintering sites to aquatic breeding sites; *Orlowski, 2007*; *Semlitsch, 2008*; *Hartel et al., 2009*; *Andrews, Nanjappa & Riley, 2015*), relatively slow movement (*Carr & Fahrig, 2001*; *Puky, 2005*; *Garrah, 2012*; *Hamer, Langton & Lesbarrères, 2015*), and the tendency to remain immobile in response to oncoming vehicles (*Mazerolle, Huot & Gravel, 2005*; *Rytwinski & Fahrig, 2012*; *Lima et al., 2015*), which make them more prone to roadkill events than most other vertebrate species when crossing roads (*Hels & Buchwald, 2001*; *Gryz & Krauze, 2008*; *Matos, Sillero & Argaña, 2012*; *Alroy, 2015*; *Heigl et al., 2017*). Additionally, because of their small body size, they are difficult for drivers to see, which increases the number of collision events (*Andrews, Nanjappa & Riley, 2015*; *Arevalo et al., 2017*). Hence, the extensive amounts of road mortality resulting from the pervasiveness of roads have prompted road planners and ecologists to implement mitigation measures to offset the negative effects that roads have on wildlife (*Costa, Ascensao & Bager, 2015*).

In recent years, the global road system has been constantly expending, and it is expected that worldwide, at least 25 million km of new roads will be built by 2050 (*Laurance et al., 2014*). For example, by the end of 2016, the total extent of China's road network reached 4.7 million km, which was a 2.5% increase from the previous year (*National Bureau of Statistics of China, 2016*). As mentioned above, road mortality has increased with the expansion of road networks and is one of several factors leading to amphibian decline (*Fahrig et al., 1995*; *Gibbs & Shriver, 2005*; *Coelho et al., 2012*; *Cosentino et al., 2014*). In addition, traffic volume is an important issue in determining the impacts of roads (*Gibbs & Shriver, 2005*; *Gu et al., 2011*; *Grilo, Ferreira & Revilla, 2015*; *Cunnington et al., 2014*; *Hamer, Langton & Lesbarrères, 2015*). Previous studies suggested that roadkill rates were positively associated with traffic intensity (*Fahrig et al., 1995*; *Gibbs & Shriver, 2002*). However, few researchers pay attention to the amphibian's roadkill in China (*Wang et al., 2013*).

In fact, a considerable number of studies have described the effects of roads on amphibians, but there are still some unresolved problems. First, many roadkill studies have focused on high-grade highways (e.g., expressways (lanes: 4–8, the average annual daily traffic (AADT): 25,000–100,000), first-class highways (lanes: 4–6, AADT: 15,000–55,000) and second-class highways (lanes: 2, AADT: 5,000–15,000); *Zhao & Deng, 2012*; *Hobday & Minstrell, 2008*; *Do Nascimento Pereira, Calabuig & Wachlevski, 2018*), with far fewer being conducted on low-grade highways (e.g., third-class highways (lanes: 2, AADT: 2,000–6,000) and fourth-class highways (lanes: 2, AADT $\leq$ 2,000 or one-lane, AADT $\leq$ 400)) and rural roads (*Gibbs & Shriver, 2005*; *Orlowski, 2007*; *Sillero, 2008*; *Matos, Sillero & Argaña, 2012*). However, the density of the road network in China reached 48.9 km per 100 square km by the end of 2016, and the mileage of highways classified as fourth-class highways or above was approximately 4.2 million km. Moreover, third-class and fourth-class highways accounted for 77.2% of the total road mileage within China (*National Bureau of Statistics of China, 2016*). With such a wide of third- and fourth-class highways it is imperative to understand the impacts that these types of road

have on wildlife. Second, roadkill studies disproportionally focus on large animals (*Biggs et al., 2004*; *Langen, Ogden & Schwarting, 2009*) due to the impact that these collisions on human safety, however it is also imperative we understand the impacts of roads on small vertebrates and the resulting ecological ramifications (*Andrews, Nanjappa & Riley, 2015*). For example, amphibians, which are considered one of the groups most vulnerable to road mortality, tend to suffer from direct mortality due to vehicle collisions more than do other vertebrates (*Goldingay & Taylor, 2006*; *Coelho et al., 2012*; *Franch et al., 2015*). Third, the rapid development of ecotourism can provide close access to Chinese reserves (*Zhou et al., 2013*). Unfortunately, increased ecotourism is accompanied by increased traffic volume, which has a destructive effect on amphibians. Thus, it is difficult to determine how to balance the public needs for increased road networks with the need to reduce amphibian roadkill. Finally, although there are a number of studies on assessment of the influences on amphibians during certain traffic volume periods (*Mazerolle, 2004*; *Sutherland, Dunning & Baker, 2010*; *Arevalo et al., 2017*), few similar studies have been performed in China. For instance, *Hels & Buchwald (2001)* conducted a study on the peninsula of Djursland, where they examined the diurnal variations in the amphibian fraction of roadkill and found that amphibian road mortality reached its highest peak after sunset, which may be attributable to their activity patterns.

Given these deficiencies, we conducted an amphibian roadkill survey on a low-grade road. The objectives of this study were (1) to investigate which species are the most vulnerable to vehicle collisions, (2) to explore the temporal and spatial roadkill distribution patterns for the surveyed amphibians, and (3) to identify the factors that contribute to amphibian road mortality. Based on our study, we can also provide targeted information (identified hot moments and hotspots of amphibian roadkill) for implementing mitigation measures through the temporal and spatial distribution patterns in amphibian mortality.

## MATERIALS AND METHODS

### Ethics statement

Our research was conducted in compliance with laws and ethical standards of China. All animal procedures were approved by the Animal Care and Use Committee of the Chengdu Institute of Biology, Chinese Academy of Sciences (CIB2015003). All field work with animals was conducted according to relevant national and international guidelines. Chengdu Institute of Biology issued permit number CIB201502 for the field work.

### Study area

We conducted this study in the Wanglang NNR (Fig. 1), which is located in Pingwu County of Sichuan Province in south-western China (103°55′–104°10′E, 32°49′–33°02′N). This reserve was established mainly for the protection of giant panda and other rare wild animals as well as of their habitat. The Wanglang NNR is one of the four earliest nature reverses established for panda in China. The Wanglang NNR is 322.97 square km in area and ranges in elevation from 2,300 to 4,980 m. The local climate is semi-humid,

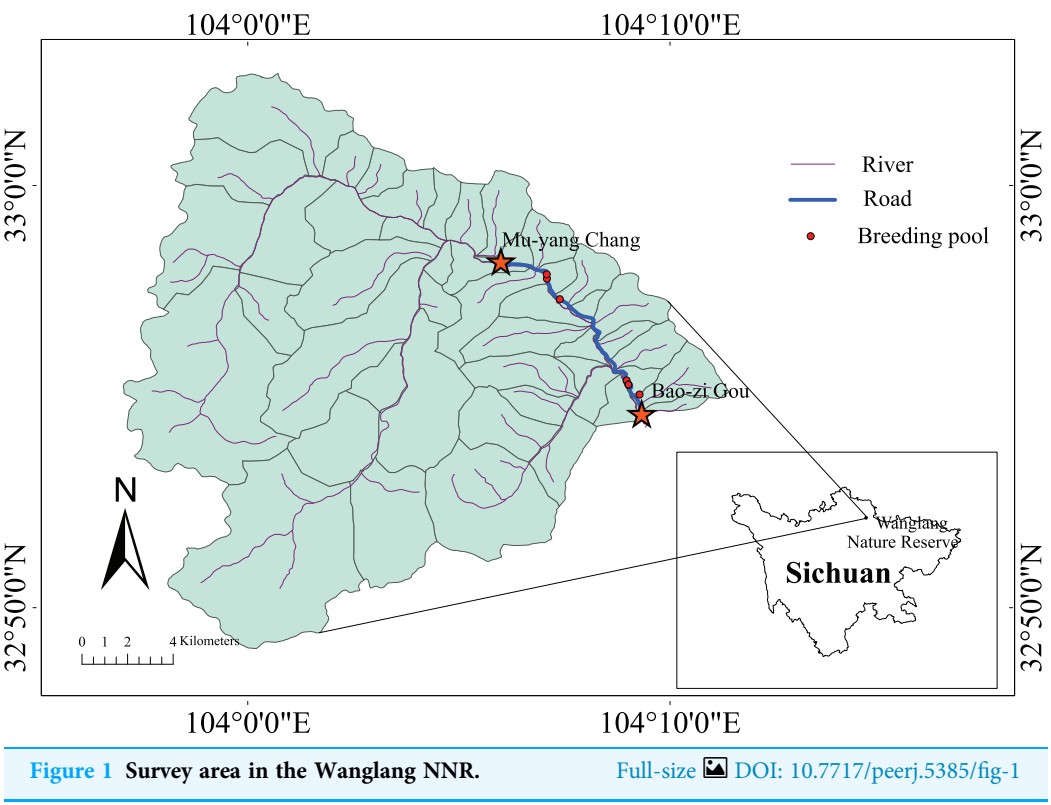

**Figure 1 Survey area in the Wanglang NNR.**

with an annual mean temperature ranging from 2.5 to 2.9 °C, and an annual mean rainfall of 859.9 mm. The rainfall is mainly concentrated in May, June, and July.

## Survey route

The survey route was approximately 10 km of a two-lane asphalt road (fourth-class highway) that was 5.5 m wide. The road was constructed in 2010 and traverses the Wanglang NNR, from the Mu-yang Chang protection station (104.10185°E, 32.96914°N) to the Bao-zi Gou protection station (104.15619°E, 32.90912°N). There are six breeding pools along the survey route. We monitored amphibian roadkill from April to October in 2017 and during June and August in 2015 and 2016. The vegetation near the survey road consisted of a secondary broad-leaved forest dominated by *Abies fabri*, *Betula albosinensis*, and *Hippophae rhamnoides*.

## Study animals

There are five amphibian species in the Wanglang NNR, including one salamander (*Batrachuperus tibetanus*), one frog (*R. chensinensis*), and three toads (*B. andrewsi*, *Oreolalax chuanbeiensis,* and *Scutiger boulengeri*); however, we found only two species, *B. andrewsi* and *R. chensinensis*, that were vulnerable to road mortality. The other species are alpine species that are often found in mountain streams and were never encountered as roadkill along the survey road. The survey road runs through the habitats of *B. andrewsi* and *R. chensinensis*, which may explain why it poses a threat to their populations. In the study area, *B. andrewsi* hibernates from late October to mid-March and

breeds in late March (*Fei et al., 2009a*), while *R. chensinensis* migrates to breeding pools in early March and begins to hibernate in late September (*Fei et al., 2009b*). During the migration season, these species cross the road to reach the breeding ponds, hibernation sites or foraging sites. Road mortality may occur during this period. Therefore, *B. andrewsi* and *R. chensinensis* were chosen as sample animals.

## Roadkill surveys

Roadkill surveys were randomly conducted on seven consecutive days per month in the Wanglang NNR, considering that most carcasses probably remain on the road for 24 h (*Santos, Carvalho & Mira, 2011*; *Brzezinski, Eliava & Zmihorski, 2012*). We conducted surveys for a total of 77 days over 11 months. The investigation lasted from April to October in 2017 for identifying the hot moments and during June and August in 2015, 2016 and 2017 for identifying the hotspots. Road monitoring was conducted by two observers who walked along the road from 08:00 to 11:30 h. Both sides of the road were carefully checked to avoid underestimation of road mortality. All carcasses of amphibians were counted, identified and marked with white foam tape to avoid double-counting. Geographic coordinates were obtained with a handheld receiver (Garmin GPS 60), and carcasses were photographed with a Sony DSC-RX100.

## Data analyses

Kruskal–Wallis tests were used to determine the differences in road casualties over the 3 years (including only June and August). To determine the fine-scale spatial distribution of amphibian roadkill, the 10 km road was divided into a grid of 0.004 degree boxes, and each of which was approximately 445 square m ($21 \times 21$ m). To identify the hotspots of amphibian road mortality, these boxes were divided into five levels of amphibian roadkills. Boxes in the first two levels, with more than 11 roadkills, were defined as the hotspots. The environmental factors related to amphibian roadkill included mean daytime temperature (DT, °C), mean night-time temperature (NT, °C), mean daytime humidity (DH, %) and mean night-time humidity (NH, %). Daytime was defined as the time between sunrise and sunset, while night-time was defined as the time between sunset and sunrise (https://avcams.faa.gov/sunrise_sunset.php). The data were obtained with a TP-2200 temperature and humidity data logger. We obtained daily precipitation (DP, mm) from the local meteorological station. To calculate the mean daily traffic volume (MDT) and mean nightly traffic volume (MNT), we checked the vehicle volume via a vehicle monitoring device in the Wanglang NNR.

   A variance inflation factor (VIF) was used to determine the factors that were collinear and should be removed before subsequent analyses (*Kutner, Nachtsheim & Neter, 2004*). As a rule of thumb, a VIF value above 4 was considered to represent problematic multicollinearity (*Kabacoff, 2010*; *Timme et al., 2017*). Therefore, we removed one of the two tested factors if the VIF was greater than 4. To identify a set of environmental factors that did not have collinearity, we recalculated the VIF values and repeated the process until all VIF values were less than 4. We began with a generalized linear model (GLM) using a Poisson distribution. Therefore, we needed to check whether

over-dispersion existed in the counts of roadkill. We conducted a Chi-squared test to compare the death data with the standard Poisson distribution for the same mean. We also tested for the model's goodness of fit via a Chi-square test based on the residual deviance and degrees of freedom. Next, we built a GLM by using the factors without multicollinearity. A GLM was performed to determine the environmental factors that were related to the daily roadkill events.

To identify the optimal model, we used a stepwise backward selection method using the Akaike information criterion (AIC; *Akaike, 1974*; *Sakamoto, Ishiguro & Kitagawa, 1986*). We ranked the models separately based on the AIC and then used the AIC values to determine the best model among all the submodels. All analyses were performed in the R statistical environment version 3.4.2 (*R Development Core Team RC, 2016*).

## RESULTS

### Species of amphibian roadkill

A total of 298 individuals were examined during the survey, including 254 toads (*B. andrewsi*) and 44 frogs (*R. chensinensis*). *B. andrewsi* was the most common amphibian species killed on the road, accounting for 85.2%, followed by *R. chensinensis* (14.8%).

### Temporal patterns of roadkill

We conducted a total of 49 road surveys (each over the same 10 km) over 490 km from April to October of 2017 and found 88 amphibian roadkills, with *B. andrewsi* accounting for the largest proportion (78.4%). The monthly numbers of amphibian roadkill events from April to October in 2017 were not homogenous. The peak amphibian roadkill rate was observed in April, when amphibians were observed to move from the breeding pools to the adjacent forest or overwintering sites. By contrast, October exhibited the lowest roadkill rates. The roadkill rate of *B. andrewsi* was highest in June and July. The rate of *R. chensinensis* roadkill was highest in April, but there were few roadkill events recorded in the surveys during other months (Fig. 2).

Kruskal–Wallis tests showed no differences in mortality within years for both species, *B. andrewsi*: $x^2 = 4.973$, d$f = 2$, $P = 0.083$, and *R. chensinensis*: $x^2 = 5.571$, d$f = 2$, $P = 0.062$. However, the results showed downward trends in the mortalities of *B. andrewsi* and *R. chensinensis* in June and August over the 3 years. *B. andrewsi* was recorded at an average rate of 0.51 ± 0.12 individuals/km/day, and *R. chensinensis* was recorded at an average rate of 0.07 ± 0.02 individuals/km/day (Fig. 3).

### Spatial patterns of roadkill

We drew a 'heatmap' based on the roadkill data that were collected over 6 months (June and August from 2015 to 2017). The spatial distribution of amphibian roadkill was not randomly distributed on the road; instead, we found obvious spatial patterns of roadkill aggregations. There were hotspots that were concentrated in specific road locations with high numbers of roadkilled amphibians. In our study, 10 boxes with more than 11 roadkill events were identified as the hotspots of amphibian roadkill, and are shown in red and orange on the map (Fig. 4). These hotspots

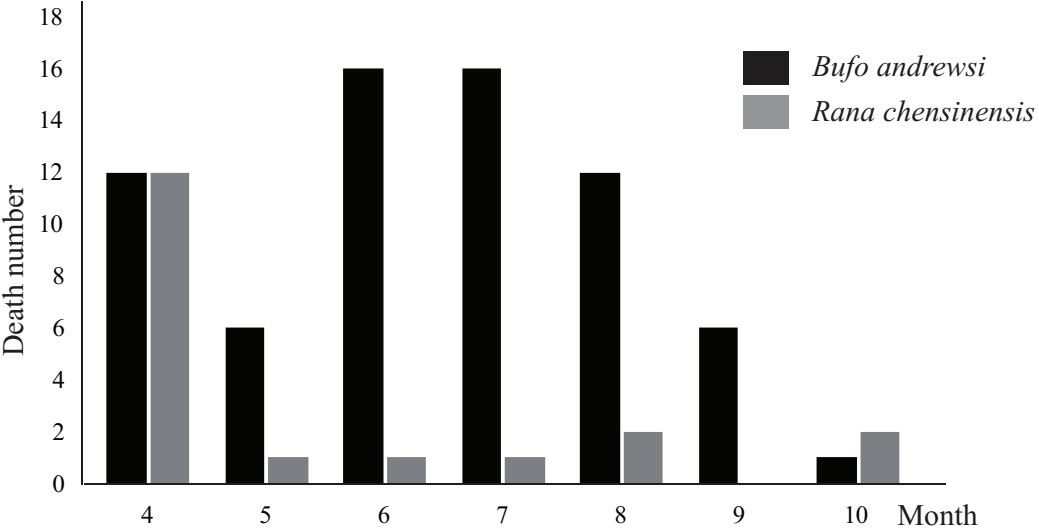

**Figure 2 The numbers of *B. andrewsi* and *R. chensinensis* deaths in different months of 2017.**

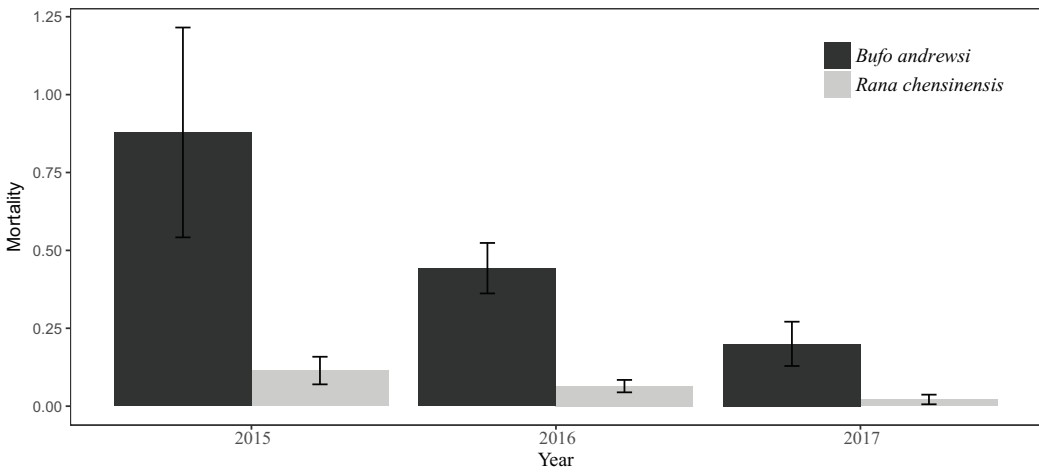

**Figure 3 Numbers of *B. andrewsi* and *R. chensinensis* deaths in June and August over the 3 years.**

(with one exception) were near breeding pools (small blue points) along the 10 km survey route (Fig. 4). However, there were fewer carcasses (green boxes) on some roads with culverts.

## Driving factors for roadkill

Data were recorded from April to October in 2017 and were used to examine whether roadkill events were associated with environmental factors. The results showed that the mean NT had the highest VIF value (VIF = 12.0241), which meant that there was a serious problem of multicollinearity between the environmental variables if we included the mean NT. Therefore, we removed the mean NT and repeated the analysis; with this change, the VIF values of the environmental factors were all less than 4, which indicated

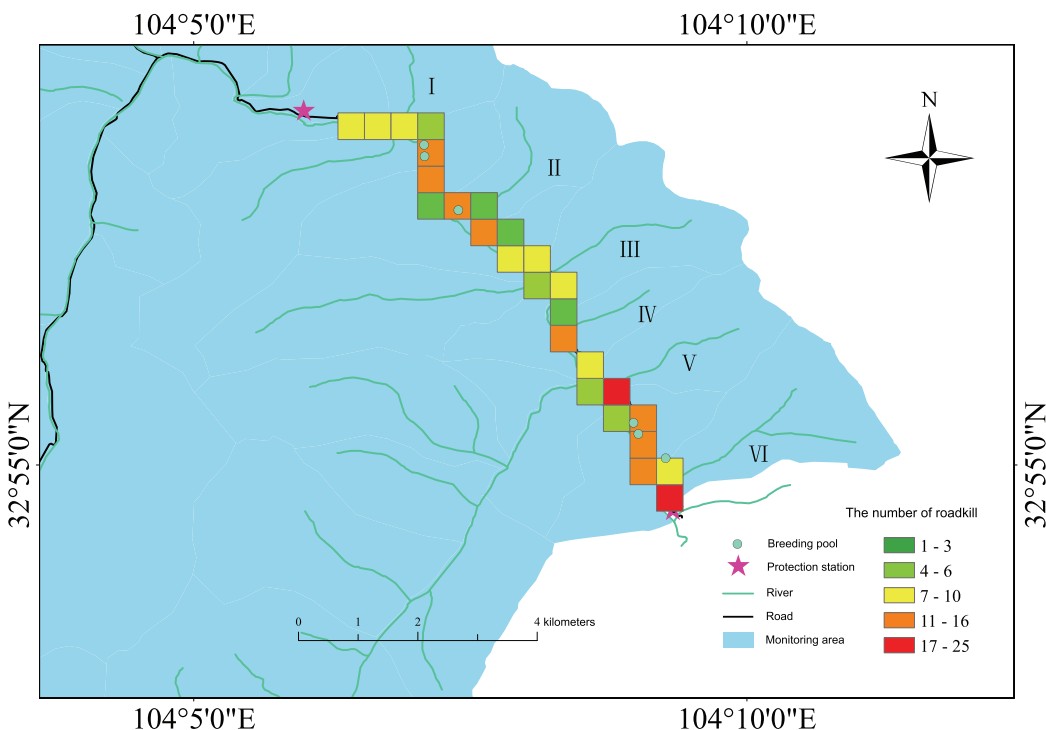

**Figure 4 The 'heatmap' of amphibian roadkill along the survey road in the Wanglang NNR.** Hotspots that are noted in red and orange along the route had the highest roadkill density. Along the surveyed road, there are six main ditches, the Nan Ditch, Qiping Ditch, Qikeshu Ditch, Jingkou Ditch, Changbai Ditch and Jiefang Ditch, and only areas II (Qiping Ditch) and IV (Qikeshu Ditch) have culverts.

**Table 1 Stepwise backward model selection for roadkill data in the Wanglang NNR.**

| Step | Deviance | Resid. df | Resid. dev | AIC |
|---|---|---|---|---|
| MNT+DH+DT+NH+DP+MDT | | 35 | 41.0793 | 152.4727 |
| MNT+DH+DT+NH+DP | 0.0090 | 36 | 41.0703 | 150.4873 |
| MNT+DH+DT+NH | 0.0249 | 37 | 41.0952 | 148.5185 |
| MNT+DH+DT | 0.0266 | 38 | 41.0686 | 147.6520 |
| MNT+DH | 0.0308 | 39 | 41.0995 | 146.1933 |
| MNT | 0.5350 | 40 | 41.6345 | 145.9125 |

that there was no multicollinearity between the factors. Therefore, the mean NT was removed from the GLM analyses. In general, the Poisson distribution can be used for fitting count data. We found that the daily roadkill events did not conform to the Poisson distribution, as the variance exceeded the mean. Thus, a negative binomial distribution was used to address the over-dispersion of the count data (*White & Bennetts, 1996*). Using the step-AIC function in the R package MASS, we computed the effects of the environmental factors by comparing the AIC values (*Burnham & Anderson, 2002*). The best model included the mean night-time traffic and mean DH (Table 1). The GLM showed that the number of amphibian roadkill events was positively related to the mean night-time traffic volume (Table 2).

**Table 2  GLM analysis of daily roadkill events and environmental factors.**

|  | Estimate | Std. error | z value | P |
|---|---|---|---|---|
| Intercept | −0.2954 | 0.3143 | −0.9400 | 0.3473 |
| MNT | 0.2924 | 0.0798 | 3.6630 | 0.0002*** |

Note:
*** Represents $P < 0.001$.

## DISCUSSION

### Magnitude of amphibian roadkill

Actual road mortality is often underestimated in studies on amphibian roadkill events (*Gibbs & Shriver, 2005*; *Langen et al., 2007*; *Elzanowski et al., 2009*; *Coelho et al., 2012*). In our study over an 11-month period from June 2015 to October 2017, we conducted roadkill monitoring surveys over 77 days and collected only 298 roadkilled individuals along the 10 km road in the Wanglang NNR. This low number of deaths might be due to uncontrollable factors. For example, we witnessed that carcasses were eaten by scavengers, such as crows, foxes and leopard cats, which may reduce the number of detected amphibian roadkills. In addition, roadkill estimations are also affected by other factors, including weather, traffic volume and research approaches (*Santos, Carvalho & Mira, 2011*; *Santos et al., 2016*). For instance, the weather affected the accuracy of the roadkill counts because it was difficult to detect the carcasses during heavy rain (*Guinard, Julliard & Barbraud, 2012*; *Beebee, 2013*). Dead amphibians became nearly completely disintegrated after traffic passed over them (*Goldingay & Taylor, 2006*), and high traffic volume might decrease the accuracy of roadkill estimations because the heavy traffic can result in rapid decomposition of the carcasses (*Mazerolle, 2004*; *Langen et al., 2007*; *Sutherland, Dunning & Baker, 2010*). Research approaches also play an important role in amphibian roadkill surveys. Counting while walking is more effective than counting while driving, since driving may cause more missed carcasses along the road (*Orlowski, 2007*; *Elzanowski et al., 2009*); however, driving can cover a larger expanse of survey routes and save time (*Langen et al., 2007*).

Our results suggested that *B. andrewsi* and *R. chensinensis* suffered serious threats due to road mortality in the Wanglang NNR. Toads were more vulnerable to traffic than frogs. The differences in road mortality between the two species might be explained by several factors, including the movement velocity of the species, detection rate, life-history traits and species abundances. First, *B. andrewsi* moves more slowly than *R. chensinensis*. Hence, *R. chensinensis* is more likely to escape collisions with cars. The low detectability of *R. chensinensis* roadkill could also be due to its small body size. Small corpses can be rapidly disintegrated by scavengers or vehicles, making it more likely that the actual number of deaths would be underestimated (*Santos, Carvalho & Mira, 2011*). The post-breeding movements of *B. andrewsi* and *R. chensinensis* were quite different, as *B. andrewsi* tended to stay near the breeding pools or forests, which may have increased the probability of encountering a road surface, thus increasing the road mortality rate. By contrast, *R. chensinensis* may stop over for several days to recuperate and

then migrate to the mountains to forage. Furthermore, species abundances also influenced the level of roadkill (*Goncalves et al., 2018*). Based on amphibian monitoring programmes conducted in the Wanglang NNR over 3 years, we found that there were more *B. andrewsi* than *R. chensinensis*.

## Roadkill hot moments

Our results showed that according to the data from 2017, the hot moments of amphibian roadkill were mainly concentrated in April. These hot moments were closely related to the life-history, phenology, and particularly migration of the species between breeding sites and hibernation sites. Other studies identified similar results and found that amphibians suffered from higher road mortality during the breeding season (*Santos et al., 2007*; *Hartel et al., 2009*). *Orlowski (2007)* studying amphibians in Lower Silesia, also detected a roadkill peak in April. By contrast, the peak rates of roadkill for *B. andrewsi* were observed in June and July, possibly because this species moves to foraging areas, which increases road mortality. However, the fewest roadkill instances were recorded in October, probably because *B. andrewsi* and *R. chensinensis* began to hibernate at that time. Most amphibians exhibited low mortality rates in the winter due to hibernation (*Seo et al., 2015*). Overall, the temporal distribution of amphibian roadkills was closely related with their breeding seasons and with rainy events (*Carvalho & Mira, 2011*).

## Roadkill hotspots

Our results showed that amphibian roadkill hotspots that occurred along the road were spatially clustered. Previous studies also demonstrated that the distribution of amphibian roadkill events was non-random along the surveyed roads (*Sillero, 2008*; *Langen, Ogden & Schwarting, 2009*; *Coelho et al., 2012*; *Seo et al., 2015*). Roadkill hotspots were predominantly associated with breeding pools. A previous study found that amphibians often used the same migratory pathways to cross the road (*Santos et al., 2007*), which would increase the probability of road mortality due to encounters with vehicles. As such, our findings support the convention that amphibians are more vulnerable to traffic mortality near water bodies (*Orlowski, 2007*; *Seo et al., 2015*; *D'Amico et al., 2015*).

## Factor driving road mortality rates

Some nature reserves in China, such as Wanglang NNR, have become popular destinations for ecotourism activities due to their unique natural landscapes and abundant animal and plant resources (*Liu, Li & Pechacek, 2013*). As a result, private tourism companies were already established in Wanglang NNR in 2015. However, the Ministry of Environmental Protection (MEP) is also aware of the negative consequences of tourism and, in 2016, introduced a regulation to control the level of travel traffic inside Wanglang NNR. This regulation reduced the number of vehicles entering the reserve, compared to the previous level of traffic. This may explain the obvious downward trend in the road mortalities of *B. andrewsi* and *R. chensinensis* in June and August from 2015 to 2017. Therefore, this suggests that the traffic volume has a substantial effect on amphibian road mortality.

Furthermore, our findings indicated that night-time traffic was a major factor contributing to amphibian roadkill, as the number of dead amphibians increased with increasing night-time traffic volume. Thus, as a more effective and targeted measure for decreasing amphibian roadkill, the MEP should reduce the night-time traffic rather than controlling the overall daily traffic. However, we also found that the number of dead amphibians may be associated with the mean DH; since AIC values were similar between the models MNT+DH and DH. Amphibian road mortality events often show a positive relation to humidity; particularly during migration, amphibians suffer higher road mortality in humid environments (*Duellman & Trueb, 1986*; *Garriga et al., 2017*). Furthermore, amphibian road mortality rates are prone to increases with increasing traffic volume (*Fahrig et al., 1995*; *Hels & Buchwald, 2001*). Previous studies have seen that the mean night-time traffic also affected amphibian road mortality (*Sutherland, Dunning & Baker, 2010*; *Coelho et al., 2012*); however, night-time traffic has rarely been specifically proposed to explain fatal collisions. In addition, previous studies have not provided detailed traffic volumes across a 24 h cycle. However, since most amphibian species are nocturnal, and most amphibian-vehicle collision occur at night (*Hels & Buchwald, 2001*), it is imperative to understand the temporal relationship between traffic flow and road mortality events. Thus, our study considered both the day-time and night-time traffic flows, instead of the overall daily traffic, and we documented that night-time traffic was a driving factor for amphibian road mortality events.

## Conservation actions

Previous studies indicate that there are many mitigation measures to reduce the amphibian roadkill (*Schmidt & Zumbach, 2008*; *Woltz, Gibbs & Ducey, 2008*; *D'Anunciação et al., 2013*). Such measures are mainly divided into two types: altering human behaviour and altering amphibian behaviour (*Beebee, 2013*). Altering human behaviour includes using warning signs to increase drivers' awareness of amphibian crossings, reducing driver speed, temporary road closures during migration and assisting amphibians across the road; altering amphibian behaviour includes constructing new breeding ponds such that road crossing is no longer necessary, as well as installing barrier fences or barrier walls in combination with underpasses (e.g., culverts or tunnels) to direct them to safe crossing sites (*Dodd, Barichivich & Smith, 2004*; *Lesbarrères, Lodé & Merilä, 2004*; *Lesbarrères et al., 2010*; *Patrick et al., 2010*; *Cunnington et al., 2014*; *Andrews, Nanjappa & Riley, 2015*; *Hamer, Langton & Lesbarrères, 2015*). These mitigation options are generally effective for most amphibians.

However, if conservation actions are to be implemented to reduce the amphibian road mortality within Wanglang NNR, then our data suggest that the most effective means to prevent road mortality would be to restrict the volume of tourist traffic at night. Overall, by reducing the amount of night-time traffic there should be a considerable decrease in amphibian road mortality, which would in turn further contribute to better more environmentally conscience ecotourism within the region. Additionally, good communication between road planners and ecologists will contribute to a better understanding of how to implement mitigation measures (*Lesbarrères & Fahrig, 2012*). Additionally, we recommend that the government should strengthen conservation

education for citizens and improve their awareness of environmental protection, which will facilitate future conservation in China.

## CONCLUSION

*B. andrewsi* and *R. chensinensis* suffered from serious threat due to road mortality in the Wanglang NNR. The night-time traffic was a major factor that contributed to amphibian roadkill. In addition, *B. andrewsi* was more vulnerable to vehicle collisions than *R. chensinensis*. Amphibian roadkill was temporally and spatially aggregated along the road in the Wanglang NNR. Roadkill hot moments were concentrated in April, June, and July; roadkill hotspots were concentrated in the road sections near the breeding pools. Based on our findings, we suggest the following mitigation measures to reduce the roadkill in the Wanglang NNR. First, based on roadkill hot moments, administrators should apply temporary traffic restraints at night during the breeding season and from June to July. Second, based on roadkill hotspots, amphibian culverts should be established in areas near breeding pools adjacent to roads, and barrier walls should be installed to guide amphibians into the culverts.

## ACKNOWLEDGEMENTS

We are grateful to Qiang Dai and Youhua Chen for comments on the manuscript. We also thank Hongliang Bu and Yuanbin Zhang for providing advice for the work. Lastly, we would like to acknowledge the editor and reviewers for their valuable comments and suggestions for improving the manuscript.

### Funding

This work was supported by the Program of Biodiversity Conservation, Ministry of Environment (No. 2111101) and the National Natural Science Foundation of China (No. 31172055 granted to Cheng Li). The funders had no role in study design, data collection and analysis, decision to publish, or preparation of the manuscript.

### Grant Disclosures

The following grant information was disclosed by the authors:
Program of Biodiversity Conservation, Ministry of Environment: 2111101.
The National Natural Science Foundation of China: 31172055.

### Competing Interests

The authors declare that they have no competing interests.

### Author Contributions

- Wenyan Zhang performed the experiments, analysed the data, contributed reagents/materials/analysis tools, prepared figures and/or tables, authored or reviewed drafts of the paper, approved the final draft.

- Guocheng Shu conceived and designed the experiments, analysed the data, contributed reagents/materials/analysis tools, prepared figures and/or tables, authored or reviewed drafts of the paper, approved the final draft.
- Yulong Li analysed the data, contributed reagents/materials/analysis tools, prepared figures and/or tables, approved the final draft.
- Shan Xiong contributed reagents/materials/analysis tools, approved the final draft.
- Chunping Liang performed the experiments, approved the final draft.
- Cheng Li conceived and designed the experiments, performed the experiments, approved the final draft.

### Animal Ethics

The following information was supplied relating to ethical approvals (i.e., approving body and any reference numbers):

This observational research was approved by the Animal Care and Use Committee of the Chengdu Institute of Biology, Chinese Academy of Sciences (CIB2015003).

### Field Study Permissions

The following information was supplied relating to field study approvals (i.e., approving body and any reference numbers):

All field work was approved by Chengdu Institute of Biology, Chinese Academy of Sciences (permit number CIB201502).

### Data Availability

The raw data are provided in the Supplemental Files.

### Supplemental Information

Supplemental information for this article can be found online at http://dx.doi.org/10.7717/peerj.5385#supplemental-information.

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
