# Peer review of "Daytime driving decreases amphibian roadkill"

_PeerJ, doi:10.7717/peerj.5385_

## Round 0.1 · original submission · Minor Revisions

The comments by the reviewers will improve the paper. Please address the edits provided by the two reviewers either in the manuscript or provide a rebuttal as to why the edits were not included. I look forward to receiving your revised manuscript.

Reviewer 1 ·

Basic reporting

Overall the general writing style is acceptable, however it certainly could use some improvements. I have attached a track-changes word document to this review with suggested changes to some of the writing style and manuscript structure.

The literature is referenced well, however I strongly feel the authors would very much benefit from reading through the recent text on small animal road ecology (see Andrews, K. M., Nanjappa, P., & Riley, S. P. (Eds.). (2015). Roads and ecological infrastructure: concepts and applications for small animals. JHU Press.). Some of the grander statements made in the manuscript are not accurate, and as such need to be re-worded, removed, or at a minimum better cited (again see track-changes word .doc file).

The general format is professional, however (as mentioned above) some changes to structure and style would improve this manuscript further.

This is not a really hypothesis driven manuscript, rather it is more exploratory in nature (e.g., what amphibians are dying on roads, where, why, and how?). This is not a negative, exploratory research certainly has it place. Overall, the findings spur concepts for conservation action, which seem to be the direct purpose of the research.

Experimental design

The questions are well defined.

"This study mainly investigated four questions. : (1) Which which species are the most vulnerable to vehicle collisions? , (2) Does does road mortality have a significant impact on amphibian breeding populations? , (3) What what are the temporal and spatial road kill distribution patterns for the surveyed amphibians, and? (4) What what factors contribute to amphibian road mortality? Based on our study, we can also provide targeted information (identified hot moments and hotspots of amphibian road kill) for implementing mitigation measures through the temporal and spatial distribution patterns in amphibian mortality."

However they are rather exploratory rather than hypothesis-driven.

The methods used to collect and analysis the data all seem appropriate.

Validity of the findings

Although not particularly novel to the field, the data collected seem sound and the findings clear. The conclusions drawn are highly locally specific, but due seem to follow the general trend observed in similar studies.

Additional comments

At its core, this is an acceptable manuscript. The data collection is clear, and the analysis seem fine. I do feel that it may benefit from more of a hypothesis/prediction based approach, rather that simply an exploratory one. Also throughout the manuscript over-generalized and grandiose statements are made that are not wholly accurate. These need to be address (please see tracked-changes word document). Furthermore, some of the writing style and structure should be improved before this manuscript can be accepted for publication

Annotated reviews are not available for download in order to protect the identity of reviewers who chose to remain anonymous.

·

Basic reporting

The article from Zhang et al. is well written and the results are in general well supported. The literature is current and well used although some comparisons are likely not relevant.
Figures are useful but legend should be presented.
Raw data were included.

Experimental design

The authors have clearly defined their objectives and made their study relevant to the field of road ecology. While the study and analyses were made rigorously, some of the details surrounding the methods could be clarified. The design is also temporarily biased since the road was surveyed 14 times in 2015 and 2016 (7 in June, 7 in August each year) as opposed to 49 times in 2017 (7 times per month in 7 consecutive months). While the authors are trying to separate their results based on these two sampling results, it has some implications on their road mortality data that are not discussed.

Validity of the findings

The validity of the findings is clear and while some of the results should be addressed more, the authors present straightforward conclusions to their study. Assessing road mortality hotspots is not novel but we know little about this threat in China and in particular in Chinese National Park adding to the value and originality of this study. As I said, some of the results should be discussed more in order to fine-tune the conclusions but overall, the results are salient and do not suffer from stretched conclusions

Additional comments

Daytime driving decreases amphibian road kill
By Zhang et al.
Ref 27511v1
This manuscript represents an interesting addition to the extensive literature on road effects on wildlife, in particular associating road mortality hotspots to abiotic factors such as traffic. Roads are known to increase population mortality but most of what we know is coming from Europe or North America. By investigating road mortality in a Chinse National Nature Reserve, this study provide additional information on the detrimental effects of roads on species in this part of the world. The design and analyses are robust and my main concerns are only two-fold:
1- The design is temporarily biased since the road was surveyed 14 times in 2015 and 2016 (7 in June, 7 in August each year) as opposed to 49 times in 2017 (7 times per month in 7 consecutive months). While the authors are trying to separate their results based on these two sampling regimes, it has some implications on their road mortality data that are not discussed.
2-while not significant, there is clearly a decrease in mortality across the 3 sampling years, a fact that is not discussed at all.
3-I suggest that the objective to assess impact on populations be removed as the method does not allow for confidently address this

Additionally, I have some minor editorial suggestions that I have added on the manuscript itself.

Best regards,
David Lesbarrères

---

## Round 0.2 · accepted · Accept

Thank you for clearly addressing and incorporating the reviewer comments. This will be a good addition to the road kill literature.

#